# Optical Absorption in Tilted Geometries as an Indirect Measurement of Longitudinal Plasma Waves in Layered Cuprates

**DOI:** 10.3390/nano14121021

**Published:** 2024-06-13

**Authors:** Niccolò Sellati, Jacopo Fiore, Claudio Castellani, Lara Benfatto

**Affiliations:** Department of Physics and ISC-CNR, “Sapienza” University of Rome, P.le Aldo Moro 5, 00185 Rome, Italy; niccolo.sellati@uniroma1.it (N.S.); jacopo.fiore@uniroma1.it (J.F.); claudio.castellani@roma1.infn.it (C.C.)

**Keywords:** layered superconductors, far-infrared conductivity, anisotropic response, Josephson plasmons, dielectric tensor

## Abstract

Electromagnetic waves propagating in a layered superconductor with arbitrary momentum, with respect to the main crystallographic directions, exhibit an unavoidable mixing between longitudinal and transverse degrees of freedom. Here we show that this basic physical mechanism explains the emergence of a well-defined absorption peak in the in-plane optical conductivity when light propagates at small tilting angles relative to the stacking direction in layered cuprates. More specifically, we show that this peak, often interpreted as a spurious leakage of the *c*-axis Josephson plasmon, is instead a signature of the true longitudinal plasma mode occurring at larger momenta. By combining a classical approach based on Maxwell’s equations with a full quantum derivation of the plasma modes based on modeling the superconducting phase degrees of freedom, we provide an analytical expression for the absorption peak as a function of the tilting angle and light polarization. We suggest that an all-optical measurement in tilted geometry can be used as an alternative way to access plasma-wave dispersion, usually measured by means of large-momenta scattering techniques like resonant inelastic X-ray scattering (RIXS) or electron energy loss spectroscopy (EELS).

## 1. Introduction

In superconductors, the breaking of the continuous gauge symmetry below the superconducting (SC) critical temperature is accompanied by the emergence of two collective modes, associated with the amplitude (Higgs) or phase (Goldstone) fluctuation of the complex SC order parameter, whose absolute value at equilibrium defines the spectral gap for single-particle excitations [1]. While the former is a massive excitation, the latter is massless at a long wavelength, reflecting the infinity of possible ground states connected by a global change of the order-parameter phase. Nonetheless, the coupling of the SC phase to the electron density is directly affected by long-range Coulomb interactions between charged electrons. This effect moves the phase mode to the plasma energy scale [2], which is usually much larger than the spectral gap. As a consequence, optical signatures at the plasma energy scale, i.e., at the zero of the dielectric function, are usually unaffected by the SC transition. A rather different phenomenology is instead observed in anisotropically layered superconductors, i.e., systems where the pairing mainly occurs within planes stacked along the *c* direction, and the SC order is established below $T_{c}$ thanks to a weak Josephson-like inter-plane interaction. The hallmark of this category is represented by high-temperature cuprates [3], where the marked anisotropy has been experimentally proven by different optical probes, starting from linear optics, which measures two well-separated energy scales for the plasma modes at long wavelengths for electric fields propagating in the CuO_2_ planes or perpendicular to them. In these systems, the incoherent quasiparticle hopping along the stacking direction makes the *c*-axis response badly metallic: in contrast, below $T_{c}$, the opening of a sizable spectral gap along with the weak inter-layer pair hopping leaves a rather sharp SC plasma edge at a frequency $\omega_{c}$ of a few THz in the optical reflectivity, which clearly testifies to the emergence of a well-defined SC Josephson plasmon. Even though this feature has been experimentally observed already in the late 1990s [4,5,6,7,8,9], renewed interest in the physics of Josephson plasmons has recently emerged. Such interest has been triggered both by the applications to nano-plasmonic [10,11] and by the role played by Josephson plasmons in non-linear THz spectroscopy [12,13,14,15,16,17]. In both cases, it becomes theoretically relevant to understand the momentum dependence of the plasmon dispersion at generic momentum, i.e., not along the main crystallographic axes. In this configuration, one immediately realizes that the anisotropy leads to a non-trivial response of the system, due to the fact that the current induced by the external electric field is no longer parallel to the field itself. As extensively discussed in Refs. [18,19,20], this mechanism leads to a mixing of the longitudinal and transverse responses inside the material, making the distinction between plasmons and polaritons blurred at momenta smaller than a scale $\bar{k}∼\sqrt{\omega_{ab}^{2}-\omega_{c}^{2}}/c$ set by the anisotropy between in-plane $\omega_{ab}$ and out-of-plane $\omega_{c}$ plasma frequencies. Since usually $\omega_{ab}\gg \omega_{c}$, the effect is relevant for non-linear Josephson plasmonics in the THz regime [21,22,23,24], but does not affect, e.g., the measurements of plasmons in RIXS [25,26,27,28,29] or EELS [30,31,32], which usually measure momenta in a fraction of the Brillouin zone. In the present manuscript, we investigate an additional consequence of the above-mentioned mixing, showing how even linear optics can be used to disentangle the longitudinal-transverse mixing in a reflection or transmission geometry, which highlights the emergence inside the material of a longitudinal response induced by an external transverse electromagnetic wave. The effect manifests as an absorption peak at a scale near $\omega_{c}$ for an electromagnetic wave traveling at a small angle with respect to the *c* direction. This feature has been measured in the past in different samples of electron-doped cuprates [33,34,35,36] below $T_{c}$, and it has often been interpreted as a leakage of the *c*-axis plasmon into the in-plane response [37]. Even more interestingly, the peak position has been shown to change by varying the wave polarization in the plane of incidence, considerably challenging the interpretation of the results. Here, we provide a full theoretical description of the microscopic mechanism behind the anomalous absorption peak, and we show that it is a direct consequence of the plasmon–polariton mixing in an anisotropically layered superconductor. We argue that this effect can be used to indirectly probe the plasmon dispersion that usually appears in RIXS and EELS experiments at much larger momenta and, by changing the light polarization, to extract the in-plane and out-of-plane plasma frequencies. Our findings are benchmarked against existing experimental data for cuprates. On a more general ground, our results offer a novel perspective on the possibility of accessing collective polariton modes in complex materials by properly engineering optical measurements.

## 2. Materials and Methods

We compute the optical conductivity for an anisotropic uniaxially layered superconductor as a response to an external electromagnetic wave traveling at a finite angle with respect to the stacking direction. We first address the problem within the standard approach of Maxwell’s equations, leading to a general expression of the response at the finite angle as a function of the conductivity tensor along the main axes. To interpret the results in the case of a layered superconductor, and to make a connection with the generalized plasma modes discussed in Refs. [18,19,20], we address the same problem within a full-quantum path-integral formalism. In superconductors, plasma modes can be studied via the fluctuations θ of the phase of the SC order parameter. By coupling the electromagnetic field A to the SC phase via the minimal-coupling substitution ∇θ→∇θ+2eA/c, we can take into account electromagnetic interactions by properly including retardation effects, responsible for the longitudinal/transverse mixing at low momenta. By introducing appropriate gauge-invariant fields ψ=∇θ+2eA/c and after integrating out the matter degrees of freedom represented by θ, we obtain directly the transverse dielectric tensor, which assumes a simple expression in terms of the plasma modes of the systems and their large-momentum limit. Finally, we discuss the experimental configuration that makes it possible to measure the plasma-mode dispersion and the in-plane and out-of-plane plasma frequencies by changing the light polarization. By computing the Fresnel conditions in the case of samples grown at a tilted angle, we demonstrate that our analytical results provide a quantitative estimate of the measured response. Conversely, the effect does not appear for non-tilted samples when the wave vector comes with a finite angle with respect to the stacking direction.

## 3. Results

### 3.1. Anisotropic Linear Response of Layered System

As discussed in the introduction, several experiments in electron-doped cuprates [33,34,35,36] have shown the emergence of a peak in the in-plane conductivity below $T_{c}$ at a frequency close to that of the out-of-plane plasma edge, with the position shifting when the light polarization is changed. This peak is often interpreted as a spurious effect due to the leakage of the *c*-axis plasmon into the in-plane response [37], and light polarization is used to remove the effect [36]. However, in Ref. [35], the problem was investigated in detail by intentionally growing a sample with the stacking direction tilted with respect to the light wave vector, and a preliminary interpretation related to such a tilted geometry has been provided. Here, we will follow the same reasoning and study the response for a propagating wave vector at a tilted angle with respect to the stacking direction. To fix the notation, in the following, we will use the convention where the SC sheets are parallel to the ab-plane and stacked along the *c*-axis. We then assume, without loss of generality, that the momentum k of the propagating wave is along the ac-plane ($k_{b}=0$). The angle between k and the *c*-axis is denoted as η and the angle between the transverse current and the *b*-axis is denoted as $\phi_{J}$ (see Figure 1 for the notation followed in this manuscript). Although discussing the Fresnel conditions at the sample/air boundary in such tilted geometries is not straightforward, we will postpone this analysis to the last section, and we will focus here on the behavior inside the sample. We are interested in determining the measured conductivity, defined as the ratio between the current J induced in the field direction and the modulus of the electric field E itself.

Because of anisotropy, the charge mobility within the planes is much higher than in between stacked layers, and the current J in the material is, in general, not parallel to E, unless propagation occurs along the principal axes of the crystal (a,b,c). Indeed, in general, one can write the conductivity tensor as follows:(1)$$\left(\begin{array}{c} J_{a}\\ J_{b}\\ J_{c} \end{array}\right)=\left(\begin{array}{ccc} \sigma_{ab} & 0 & 0\\ 0 & \sigma_{ab} & 0\\ 0 & 0 & \sigma_{c} \end{array}\right)\left(\begin{array}{c} E_{a}\\ E_{b}\\ E_{c} \end{array}\right),$$
where $\sigma_{ab}$ and $\sigma_{c}$ are the in-plane and out-of-plane conductivities, respectively. In the following, we simplify the tensorial notation by writing the reference frame where a quantity is considered as its subscript, e.g., Equation (Equation 1) reads $J_{abc}={\hat{\sigma }}_{abc}E_{abc}$. If the wave propagates perpendicularly to the planes (η=0), the electric field oscillates within the SC sheets and one directly extracts $\sigma_{ab}$ from the measured transmissivity/reflectivity; analogously, with a wave propagating within the planes (η=π/2), one can measure $\sigma_{c}$. However, for a generic value of the propagation angle, the measured conductivity will be a combination of the two quantities. In other words, for k at a generic angle of η, the current J will develop both longitudinal and transverse components with respect to the momentum.

To see this explicitly, we perform a rotation of angle η around the *b*-axis to move in the reference frame (t,b,l), where *l* labels the longitudinal components and *t* labels the transverse component with respect to the momentum in the ac-plane, while preserving the second transverse component *b*. In this frame, Equation (Equation 1) transforms into $J_{tbl}={\hat{\sigma }}_{tbl}E_{tbl}$, where the conductivity tensor now reads as follows:(2)$$\begin{array}{c} {\hat{\sigma }}_{tbl}=\left(\begin{array}{ccc} \sigma_{ab}\cos^{2}\eta +\sigma_{c}\sin^{2}\eta & 0 & (\sigma_{c}-\sigma_{ab})\sin\eta \cos\eta \\ 0 & \sigma_{ab} & 0\\ (\sigma_{c}-\sigma_{ab})\sin\eta \cos\eta & 0 & \sigma_{ab}\sin^{2}\eta +\sigma_{c}\cos^{2}\eta \end{array}\right).\; \end{array}$$

Notice that the components $J_{t}$ and $J_{l}$ are coupled to both $E_{t}$ and $E_{l}$, as one expects in an anisotropic crystal, whereas the transverse $J_{b}$ component only couples to $E_{b}$. As we will detail below, what one determines experimentally is an effective conductivity defined as the ratio between the transverse current and the transverse electric field. According to Ampere’s law, $\frac{4\pi }{c}J+\frac{\varepsilon_{\infty }}{c}\frac{\partial E}{\partial t}=0$, where *c* is the light velocity and $\varepsilon_{\infty }$ is the background dielectric constant, ensuring that the current in the longitudinal direction is compensated by the displacement current. We then derive the relation $4\pi J_{l}-i\omega \varepsilon_{\infty }E_{l}=0$, which can be used to eliminate the longitudinal component and write a system that only takes the transverse components *t* and *b* into consideration, $J_{tb}={\hat{\sigma }}_{tb}E_{tb}$. The transverse conductivity tensor reads as follows:(3)$$\begin{array}{c} {\hat{\sigma }}_{tb}=\left(\begin{array}{cc} \sigma_{t} & 0\\ 0 & \sigma_{ab} \end{array}\right),\; \end{array}$$
where
(4)$$\begin{array}{c} \sigma_{t}\left(\omega ,\eta \right)=\frac{-\frac{i\omega \varepsilon_{\infty }}{4\pi }\left(\sigma_{ab}\cos^{2}\eta +\sigma_{c}\sin^{2}\eta \right)+\sigma_{ab}\sigma_{c}}{-\frac{i\omega \varepsilon_{\infty }}{4\pi }+\sigma_{ab}\sin^{2}\eta +\sigma_{c}\cos^{2}\eta }.\; \end{array}$$

For an electric field polarized along *t* ($E_{b}=0$), Equation (Equation 4) immediately gives the conductivity we are looking for, $\sigma_{t}=J_{t}/E_{t}$. This expression was first derived in Ref. [34]; its real part displays a peak with the central frequency that moves with η. To show it explicitly, we replace $\sigma_{ab}=\frac{i\omega }{4\pi }\frac{\varepsilon_{\infty }\omega_{ab}^{2}}{{\left(\omega +i0^{+}\right)}^{2}}$ and $\sigma_{c}=\frac{i\omega }{4\pi }\frac{\varepsilon_{\infty }\omega_{c}^{2}}{{\left(\omega +i0^{+}\right)}^{2}}$, where $\omega_{ab}$ and $\omega_{c}$ are the in-plane and out-of-plane plasma frequencies, respectively: one then immediately sees that the real part of $\sigma_{t}\left(\omega ,\eta \right)$ peaks at a frequency of $\omega_{l}\left(\eta \right)$, which reads as follows:(5)$$\begin{array}{c} \omega_{l}^{2}\left(\eta \right)=\omega_{ab}^{2}\sin^{2}\eta +\omega_{c}^{2}\cos^{2}\eta .\; \end{array}$$

As we will discuss below, $\omega_{l}$ does *not* define a plasma mode of the system; this can be immediately understood within a classical approach, by writing explicitly the dielectric function corresponding to the conductivity (Equation 4). By using $\sigma_{ab}=-\frac{i\omega }{4\pi }\left(\varepsilon_{ab}-\varepsilon_{\infty }\right)$ and $\sigma_{c}=-\frac{i\omega }{4\pi }\left(\varepsilon_{c}-\varepsilon_{\infty }\right)$, we can write the in-plane $\varepsilon_{ab}$ and out-of-plane $\varepsilon_{c}$ dielectric functions of the SC system as follows:(6)$$\begin{array}{c} \varepsilon_{ab}\left(\omega \right)=\varepsilon_{\infty }\left(1-\frac{\omega_{ab}^{2}}{{\left(\omega +i0^{+}\right)}^{2}}\right),\; \end{array}$$
and
(7)$$\begin{array}{c} \varepsilon_{c}\left(\omega \right)=\varepsilon_{\infty }\left(1-\frac{\omega_{c}^{2}}{{\left(\omega +i0^{+}\right)}^{2}}\right).\; \end{array}$$

Thus, Equation (Equation 4) can be recast as $\sigma_{t}=-\frac{i\omega }{4\pi }\left(\varepsilon_{t}-\varepsilon_{\infty }\right)$, where
(8)$$\begin{array}{c} \varepsilon_{t}\left(\omega ,\eta \right)=\varepsilon_{\infty }\frac{\left({\left(\omega +i0^{+}\right)}^{2}-\omega_{ab}^{2}\right)\left({\left(\omega +i0^{+}\right)}^{2}-\omega_{c}^{2}\right)}{{\left(\omega +i0^{+}\right)}^{2}\left({\left(\omega +i0^{+}\right)}^{2}-\omega_{l}^{2}\left(\eta \right)\right)}.\; \end{array}$$

In Equation (Equation 8), the frequency $\omega_{l}\left(\eta \right)$ in Equation (Equation 5) appears as a divergence of the dielectric function, while the plasma frequencies in the long-wavelength limit appear as zeros of the dielectric function. This already proves that the scale $\omega_{l}$ does not identify a true plasma mode. However, as we will demonstrate below, it turns out that $\omega_{l}$ provides a good approximation for the *finite-momentum* longitudinal plasmon of the layered system *at large momenta*, i.e., in the momentum regions where retardation effects are no longer relevant. As a consequence, the present results show that the optical absorptive peak in the tilted geometry, which appears as a linear response in the long-wavelength limit, can be used to indirectly access the plasma-wave dispersions at large momenta. Notice that, in principle, Equation (Equation 4) is valid, in general, for any collective mode in an anisotropic uniaxial system, provided that the corresponding expressions of $\sigma_{ab}\left(\omega \right)$ and $\sigma_{c}\left(\omega \right)$ are used.

Even though these considerations solve the problem of defining a transverse conductivity at tilted angles for an electric field polarized along *t*, two main issues remain. The first one regards the connection between the frequency of the peak (Equation 5) and the real plasma modes of the anisotropic superconductor. The second point is to link these results to the measured quantity in an experiment with a generic polarization of the electric field. The first matter will be discussed in the next section using a quantum formalism based on the description of electromagnetic modes via the SC phase degree of freedom. The second issue will be the subject of the last section, where we will explicitly study the Fresnel problem for transmission/reflection through a sample grown at a tilt. Furthermore, we will discuss the dependence of the measurement on the polarization ϕ of the external incident’s electric field.

### 3.2. Linear Response of Generalized Plasma Modes

#### 3.2.1. Effective Action Description of Plasma Modes

To gain more physical insight into the results of the previous section, we will take advantage of the description of the plasma modes in the SC state obtained via the phase degrees of freedom. Indeed, as recently discussed in Refs. [18,19], this approach is both powerful and elegant in describing the interplay between longitudinal and transverse plasma waves in a layered superconductor, which leads to generalized plasma modes with mixed character at low momenta. Here, we summarize the main ingredients of the derivation, referring the reader to Refs. [18,19] and the references therein for a detailed derivation of the layered phase-only model.

Below the critical temperature $T_{c}$, the neighboring SC planes interact with a Josephson-like coupling [3,12,23,38,39,40,41] that is much weaker than the in-plane phase stiffness. Following the notation set above, we denote the in-plane superfluid stiffness by $D_{ab}$ and the out-of-plane one by $D_{c}$, and we write the Gaussian action for the phase fluctuations θ as follows [42,43,44]:(9)$$\begin{array}{c} S_{G}\left[\theta \right]=\frac{1}{8}\sum_{q}\left[\kappa_{0}\Omega_{m}^{2}+D_{ab}k_{ab}^{2}+D_{c}k_{c}^{2}\right]{\left|\theta \left(q\right)\right|}^{2},\; \end{array}$$
where $q=(i\Omega_{m},k)$ is the imaginary-time 4-momentum, with $\Omega_{m}=2\pi mT$ representing the bosonic Matsubara frequencies. Here, $k_{ab}=\sqrt{k_{a}^{2}+k_{b}^{2}}$ and $k_{c}$ are the in-plane and out-of-plane momenta, respectively, and $\kappa_{0}$ is the compressibility. In the following, we will denote by ${\left|k\right|}^{2}=k_{ab}^{2}+k_{c}^{2}$. We introduce the electromagnetic field A by performing in Equation (Equation 9) the minimal coupling substitution ikθ→ikθ+2eA/c, where −e is the charge of the electron, and we will also add the action of the free electromagnetic field [1], as follows:(10)$$\begin{array}{c} S_{e.m.}\left[A\right]=\frac{1}{8\pi c^{2}}\sum_{q}\left[\varepsilon_{\infty }\Omega_{m}^{2}{\left|A\left(q\right)\right|}^{2}+c^{2}{\left|k\times A\left(q\right)\right|}^{2}\right], \end{array}$$

Both the minimal coupling substitution and Equation (Equation 10) are written in the Weyl gauge where the scalar potential is zero. We then recast the coupling between the phase fluctuations and the electromagnetic field by performing the substitution:(11)$$\begin{array}{c} \psi \left(q\right)=ik\theta \left(q\right)+\frac{2e}{c}A\left(q\right). \end{array}$$

These gauge-invariant fields provide a full description of the plasma modes once the phase fluctuations are integrated out [18,19]. To provide simple analytical expressions, in the following, we consider the limit for infinite compressibility. This is a good approximation in single-layer cuprates, as the effects of finite compressibility on the properties of the generalized plasma modes are negligible at small momenta [18]. Interestingly, these effects are actually crucial when studying the optical absorptive peak of bilayer superconductors [4,5,6,7,8,9,13,15], whose central frequency is significantly influenced by the compressibility [45,46] due to the capacitive coupling between planes surviving at vanishing momentum [19]. In the basis $\psi_{abc}={\left(\begin{array}{ccc} \psi_{a} & \psi_{b} & \psi_{c} \end{array}\right)}^{T}$, the action of the systems after the integration of θ reads as follows:(12)$$\begin{array}{c} S\left[\psi_{abc}\right]=\frac{1}{32\pi e^{2}}\sum_{q}\psi_{abc}^{T}\left(-q\right)\left(\begin{array}{ccc} \Omega_{m}^{2}\varepsilon_{ab}+c^{2}k_{c}^{2} & 0 & -c^{2}k_{a}k_{c}\\ 0 & \Omega_{m}^{2}\varepsilon_{ab}+c^{2}{\left|k\right|}^{2} & 0\\ -c^{2}k_{a}k_{c} & 0 & \Omega_{m}^{2}\varepsilon_{c}+c^{2}k_{a}^{2} \end{array}\right)\psi_{abc}\left(q\right), \end{array}$$
where we set the in-plane momentum along the *a*-direction ($k_{b}=0$) without loss of generality, such that $\psi_{b}$ is decoupled, in full analogy with the case of Equation (Equation 2). In the action we have defined, using the Matsubara formalism, the in-plane dielectric function is as follows:(13)$$\begin{array}{c} \varepsilon_{ab}\left(i\Omega_{m}\right)=\varepsilon_{\infty }\left(1+\frac{\omega_{ab}^{2}}{\Omega_{m}^{2}}\right), \end{array}$$
and the out-of-plane dielectric function is as follows:(14)$$\begin{array}{c} \varepsilon_{c}\left(i\Omega_{m}\right)=\varepsilon_{\infty }\left(1+\frac{\omega_{c}^{2}}{\Omega_{m}^{2}}\right), \end{array}$$
where the plasma frequencies are linked to the in-plane and out-of-plane superfluid stiffness, $\omega_{ab}^{2}=4\pi e^{2}D_{ab}/\varepsilon_{\infty }$ and $\omega_{c}^{2}=4\pi e^{2}D_{c}/\varepsilon_{\infty }$, respectively. Indeed, these go back to Equations (Equation 6) and (Equation 7) once the analytic continuation $i\Omega_{m}\to \omega +i0^{+}$ is performed. Notice that the dielectric tensor is diagonal in the basis $\psi_{abc}$, as (a,b,c) is the reference frame of the principal axes of the crystal. By their definition in Equation (Equation 11), the gauge-invariant fields are formally proportional to the currents; thus, within the effective-action framework, we can apply the same procedure used above in the classical approach, i.e., a change of the reference frame to describe a transverse dielectric tensor. We, thus, perform a rotation around the *b*-axis that combines the $\psi_{a}$ and $\psi_{c}$ components into transverse $\psi_{t}$ and longitudinal $\psi_{l}$ components with respect to the momentum k. The matrix that performs the change of basis $\psi_{abc}\to \psi_{tbl}={\left(\begin{array}{ccc} \psi_{t} & \psi_{b} & \psi_{l} \end{array}\right)}^{T}$ reads as follows:(15)$$\begin{array}{c} U=\left(\begin{array}{ccc} k_{c}/\left|k\right| & 0 & k_{a}/\left|k\right|\\ 0 & 1 & 0\\ -k_{a}/\left|k\right| & 0 & k_{c}/\left|k\right| \end{array}\right), \end{array}$$
and Equation (Equation 12) transforms in this basis as follows:(16)$$\begin{array}{c} S\left[\psi_{tbl}\right]=\frac{1}{32\pi e^{2}}\sum_{q}\psi_{tbl}^{T}\left(-q\right)D_{tbl}^{-1}\psi_{tbl}\left(q\right), \end{array}$$
where the matrix of the coefficients reads
(17)$$\begin{array}{c} D_{tbl}^{-1}=\left(\begin{array}{ccc} \Omega_{m}^{2}\left(\varepsilon_{ab}k_{c}^{2}+\varepsilon_{c}k_{a}^{2}\right)/{\left|k\right|}^{2}+c^{2}{\left|k\right|}^{2} & 0 & \Omega_{m}^{2}\left(\varepsilon_{c}-\varepsilon_{ab}\right)k_{a}k_{c}/{\left|k\right|}^{2}\\ 0 & \Omega_{m}^{2}\varepsilon_{ab}+c^{2}{\left|k\right|}^{2} & 0\\ \Omega_{m}^{2}\left(\varepsilon_{c}-\varepsilon_{ab}\right)k_{a}k_{c}/{\left|k\right|}^{2} & 0 & \Omega_{m}^{2}\left(\varepsilon_{ab}k_{a}^{2}+\varepsilon_{c}k_{c}^{2}\right)/{\left|k\right|}^{2} \end{array}\right). \end{array}$$

Before moving forward and studying the linear response, we will provide a brief review of the generalized plasma modes that Equation (Equation 16) describes. This review will be useful in the following to provide a physical interpretation of the finite-frequency peak in the real part of the conductivity. The action identifies two longitudinal-transverse mixed modes and one decoupled purely transverse mode along the *b*-direction. The former ones cannot be studied separately, as the anisotropy of layered superconductors is such that the $\psi_{t}$ and $\psi_{l}$ components are coupled for the generic directions of the momenta, i.e., the off-diagonal elements of Equation (Equation 17) are nonvanishing. On physical grounds, this is a manifestation of retardation effects: as seen in the previous section, at a generic wave vector, the current induced in the system is not parallel to E. This induces a longitudinal electric field in the system in response to a *transverse* perturbation, making longitudinal and transverse responses unavoidably mixed. Since the displacement current scales as ∂E/∂(ct), the corrections coming from retardation effects are also named relativistic, as they vanish when c→∞. The dispersion relations of the two modes obtained from Equation (Equation 17) read as follows:(18)$$\begin{array}{c} \omega_{\pm }^{2}\left(k\right)=\frac{1}{2}\left(\omega_{ab}^{2}+\omega_{c}^{2}+\frac{c^{2}}{\varepsilon_{\infty }}{\left|k\right|}^{2}\pm \sqrt{{\left(\omega_{ab}^{2}-\omega_{c}^{2}\right)}^{2}+\frac{c^{4}}{\varepsilon_{\infty }^{2}}{\left|k\right|}^{4}-2\frac{c^{2}}{\varepsilon_{\infty }}\left(k_{a}^{2}-k_{c}^{2}\right)\left(\omega_{ab}^{2}-\omega_{c}^{2}\right)}\right), \end{array}$$

A detailed discussion of the properties of the generalized plasma modes of single-layer anisotropic superconductors can be found in Ref. [18]. Nonetheless, it is important here to stress the main physical outcomes of the present derivation. The generalized dispersions (Equation 18) describe two regular functions of the momenta that give $\omega_{+}\left(k\to 0\right)\to \omega_{ab}$ and $\omega_{-}\left(k\to 0\right)\to \omega_{c}$. For generic propagation direction η and for momenta $\left|k\right|≲\bar{k}=\sqrt{\varepsilon_{\infty }\left(\omega_{ab}^{2}-\omega_{c}^{2}\right)}/c$, these modes have mixed longitudinal/transverse character, with a degree of mixing that is maximum at η=π/4 and vanishes as one moves along the main crystallographic direction ($k_{a}=0$ or $k_{c}=0$), as one immediately realizes by the structure of the off-diagonal matrix elements of Equation (Equation 17), scaling as $k_{a}k_{c}$. Explicitly neglecting this coupling, i.e., setting the off-diagonal elements to zero, would result in having the two modes uncoupled, one of which is purely transverse and the other purely longitudinal. In this case, the dispersion relation of the latter, by definition, the plasma mode of the system, can be found by setting to zero the bottom-right element of $D_{tbl}^{-1}$:(19)$$\begin{array}{c} \varepsilon_{ab}\left(\omega \right)\frac{k_{a}^{2}}{{\left|k\right|}^{2}}+\varepsilon_{c}\left(\omega \right)\frac{k_{c}^{2}}{{\left|k\right|}^{2}}=\varepsilon_{ab}\left(\omega \right)\sin^{2}\eta +\varepsilon_{c}\left(\omega \right)\cos^{2}\eta =0, \end{array}$$
where we have performed the analytic continuation $i\Omega_{m}\to \omega +i0^{+}$ and used $k_{c}=\left|k\right|\cos\eta$ and $k_{a}=\left|k\right|\sin\eta$. Using the definitions of the dielectric functions in Equations (Equation 13) and (Equation 14), the solution of Equation (Equation 19) is exactly the frequency, i.e.,
(20)$$\omega_{l}^{2}\left(k\right)=\omega_{ab}^{2}\frac{k_{a}^{2}}{{\left|k\right|}^{2}}+\omega_{c}^{2}\frac{k_{c}^{2}}{{\left|k\right|}^{2}}\equiv \omega_{ab}^{2}\sin^{2}\eta +\omega_{c}^{2}\cos^{2}\eta ,$$
as defined in Equation (Equation 5). In addition, one can easily see from Equation (Equation 18) that in the limit c→∞, i.e., in the regime where $\bar{k}/\left|k\right|\to 0$, retardation (or relativistic) effects can be neglected; thus, one obtains the following:(21)$$\omega_{-}\left(k\right)\to \omega_{l}\left(k\right),\;\left|k\right|\gg \bar{k}=\sqrt{\varepsilon_{\infty }\left(\omega_{ab}^{2}-\omega_{c}^{2}\right)}/c.$$
In other words, the expression $\omega_{l}\left(\eta \right)$ defines the longitudinal plasmon dispersion in a layered superconductor that one obtains by neglecting retardation effects, as one usually does in the standard RPA approach where only Coulomb interactions are included [42,43,44,47,48,49,50]. We also note in passing that the limit of $\omega_{l}\left(k\right)$ for k→0 is non-regular as it depends on the direction η of the momentum. As shown above, this is not the case for the real electromagnetic mode $\omega_{-}$, which is regular at |k|=0. In Figure 2a, we show $\omega_{-}\left(k\right)$ and $\omega_{l}\left(\eta \right)$ for small values of the propagation angle: as one can see, as |k| overcomes the $\bar{k}$ scale, $\omega_{-}$ rapidly approaches the $\omega_{l}$ limit and the mode becomes longitudinal. By using realistic values of plasma frequencies in cuprates, one sees that $\bar{k}∼\mu m^{-1}$. As such, this scale is two orders of magnitude smaller than the momenta usually accessible in RIXS [25,26,27,28,29] or EELS [30,31,32] experiments, which are not sensitive to the relativistic regime and probe the plasmon dispersion given by Equation (Equation 20).

#### 3.2.2. Interpretation of the Conductivity Peak of Plasmons

From the action in Equation (Equation 16), we can perform the integration of $\psi_{l}$ and work with an action of the transverse components $\psi_{tb}={\left(\begin{array}{cc} \psi_{t} & \psi_{b} \end{array}\right)}^{T}$ only. This procedure is equivalent to using Ampere’s law as a condition to eliminate the longitudinal components; see Equation (Equation 2) and the discussion below. One is left with an action that reads as follows:(22)$$\begin{array}{c} S\left[\psi_{tb}\right]=\frac{1}{32\pi e^{2}}\sum_{q}\psi_{tb}^{T}\left(-q\right)\left(\begin{array}{cc} \Omega_{m}^{2}\varepsilon_{t}+c^{2}{\left|k\right|}^{2} & 0\\ 0 & \Omega_{m}^{2}\varepsilon_{ab}+c^{2}{\left|k\right|}^{2} \end{array}\right)\psi_{tb}\left(q\right), \end{array}$$
where
(23)$$\begin{array}{c} \varepsilon_{t}\left(i\Omega_{m},\eta \right)=\frac{\varepsilon_{ab}\varepsilon_{c}}{\varepsilon_{ab}\sin^{2}\eta +\varepsilon_{c}\cos^{2}\eta }, \end{array}$$
is a dielectric function that describes the transverse linear response of the superconductor along the *t*-axis. Indeed, by making use of the relation $\varepsilon_{\alpha }=\varepsilon_{\infty }+4\pi i\sigma_{\alpha }/\omega$ between the optical conductivity and the dielectric function along the direction α [51], one recovers $\sigma_{t}$ as in Equation (Equation 4). Remarkably, the denominator of $\varepsilon_{t}$ can be brought back to the left-hand side of the characteristic Equation (Equation 19) for the uncoupled longitudinal mode. Indeed, by using the explicit expressions in Equations (Equation 13) and (Equation 14) for the in-plane and out-of-plane dielectric functions of plasma modes, and performing the analytic continuation $i\Omega_{m}\to \omega +i0^{+}$, one can rewrite Equation (Equation 23) as $\varepsilon_{t}\left(\omega ,\eta \right)=\varepsilon_{\infty }\left(\omega^{2}-\omega_{ab}^{2}\right)\left(\omega^{2}-\omega_{c}^{2}\right)/\left[\omega^{2}\left(\omega^{2}-\omega_{l}^{2}\left(\eta \right)\right)\right]$, exactly as in Equation (Equation 8) above.

While this result has been formally obtained within Maxwell’s classical formalism in the previous section, we can now identify the energy of the peak in the *transverse* conductivity at vanishing momentum as the value of the *longitudinal* plasma mode in the high-momentum regime. This is the same regime usually probed by EELS and RIXS since $\omega_{l}\left(\eta \right)$ is a good approximation of the dispersion of the lower mode $\omega_{-}\left(k\right)$ for $\left|k\right|\gg \bar{k}$; see Equation (Equation 21).

The real part of the conductivity $\sigma_{t}$ is shown in Figure 2b, where we also introduce a finite damping parameter γ when performing the analytic continuation $i\Omega_{m}\to \omega +i\gamma$. We emphasize once more that such a peak is not a direct manifestation of the Josephson plasmon of the superconductor [33,34,35,36], which, as discussed above, for vanishing momentum is at frequency $\omega_{c}$ for every direction η. Indeed, plasma modes appear as zeroes of the dielectric function and do not lead to finite-frequency peaks in the conductivity. Instead, the absorptive peak at $\omega_{l}\left(\eta \right)$ is a manifestation of the mixing mechanism between in-plane and out-of-plane plasma modes described in the previous section, as the dielectric function in Equation (Equation 8) comes directly from the action for the coupled modes in Equation (Equation 16). On a more general ground, our derivation clarifies that a signature of a longitudinal nature appears in the transverse response whenever the longitudinal mode is coupled to the transverse one without directly participating in the detection, i.e., the degree of freedom is integrated out.

It is worth mentioning that our derivation is not restricted to electron-doped cuprates, where the peak has already been experimentally reported [33,34,35,36], but it is valid for any single-layer superconductor, like the hole-doped LSCO. We also point out that the results could be extended to bilayer superconductors like YBCO, which display two Josephson plasmons at frequencies $\omega_{c1}$ and $\omega_{c2}$. Indeed, by using the out-of-plane bilayer dielectric function [5] $\varepsilon_{c}=\varepsilon_{\infty }\left(\omega^{2}-\omega_{c1}^{2}\right)\left(\omega^{2}-\omega_{c2}^{2}\right)/\left[\omega^{2}\left(\omega^{2}-\omega_{T}^{2}\right)\right]$, with $\omega_{T}^{2}=\omega_{c1}^{2}d_{2}+\omega_{c2}^{2}d_{1}$ and $d_{1,2}$ representing the intra- and inter-bilayer spacings, one can predict two absorptive peaks in the conductivity. These peaks are centered at the high-momentum values of the dispersions of the Josephson modes [19]. The high-energy peak follows the same trend as the peak in single-layer superconductors, moving with η from $\omega_{c1}$ to $\omega_{ab}$. The low-energy one quickly moves from $\omega_{c2}$ to $\omega_{T}$ even for small values of η and does not disappear for η=π/2 [4,5,6,7,8,9,13,15]. As mentioned above, finite compressibility corrections are crucial for optical measurements in bilayer superconductors [45,46], and they must be taken into account to correctly fit the experimental data [19].

### 3.3. Fresnel Equations at Normal Incidence on a Tilted-Grown Sample

To link the results obtained in the previous sections to experiments, we must consider the measured quantity, which is the electric field transmitted or reflected through the sample relative to the incident wave, and relate it to the conductivity $\sigma_{t}$. Moreover, one might argue that due to the fact that the system is anisotropic, both angles η and $\phi_{J}$ that define the current propagation within the material differ, respectively, from $\eta_{in}$, the angle between the incident momentum of the external wave and the normal to the planes, and ϕ, the angle between the *b*-axis and the electric field that describes its polarization. To this aim, we must write the Fresnel conditions at the boundaries of the sample. In this section, we analyze the configuration where a THz pulse is at normal incidence on a thin-film layered superconductor, grown with tilted planes at a small angle η [33,34,35,36], and we show that in this case, the theoretical results can be easily related to experiments; see Appendix A.

Following the notation set above, we define the reference frame (t,b,l) such that the tb-plane corresponds to the interface and the *l*-axis is perpendicular to it, see Figure 3a. At normal incidence, $\eta_{in}=\eta$ immediately, as the momentum of the wave does not change direction when crossing the interface. Within the material, the *b*-polarized and *t*-polarized electric fields are decoupled and travel with different values of the wave vector; see Equation (Equation 16) and the discussion below. In particular, from Equation (Equation 17), the equations of motion read ${\left|k\right|}^{2}=\omega^{2}\varepsilon_{ab}/c^{2}$ for the former and ${\left|k\right|}^{2}=\omega^{2}\varepsilon_{t}/c^{2}$ for the latter [22], with $\varepsilon_{t}$ defined in Equation (Equation 23). We then impose the continuity of the tangential components of the electric fields, $E_{t}$ and $E_{b}$, and of the magnetic fields, $B_{t}$ and $B_{b}$, at interfaces l=0 and l=d, with *d* denoting the sample thickness. By solving the system set by these conditions, one finds the transmission and reflection coefficients for the *t* and *b* components of the field in the thin-film configuration, which read as follows: (24)$$\begin{array}{cc} \; & T_{t}=\frac{T_{t}T_{t}^{\prime }e^{in_{t}\omega d/c}}{1-R_{t}^{2}e^{2in_{t}\omega d/c}}, \end{array}$$(25)$$\begin{array}{cc} \; & R_{t}=\frac{R_{t}\left(1-e^{2in_{t}\omega d/c}\right)}{1-R_{t}^{2}e^{2in_{t}\omega d/c}}, \end{array}$$(26)$$\begin{array}{cc} \; & T_{b}=\frac{T_{b}T_{b}^{\prime }e^{in_{b}\omega d/c}}{1-R_{b}^{2}e^{2in_{b}\omega d/c}}, \end{array}$$(27)$$\begin{array}{cc} \; & R_{b}=\frac{R_{b}\left(1-e^{2in_{b}\omega d/c}\right)}{1-R_{b}^{2}e^{2in_{b}\omega d/c}}, \end{array}$$
where $n_{\alpha }=\sqrt{\varepsilon_{\alpha }}$ is the refractive index along the direction α, $T_{\alpha }=2/\left(1+n_{\alpha }\right)$ is the transmission coefficient going from the vacuum to the material, $T_{\alpha }^{\prime }=2n_{\alpha }/\left(1+n_{\alpha }\right)$ is—analogously—the transmission coefficient from the sample to the vacuum, and $R_{\alpha }^{2}=1-T_{\alpha }T_{\alpha }^{\prime }$ accounts for the Fabry–Perot interference within the thin film. The ratios $T_{t}/T_{b}$ and $R_{t}/R_{b}$ carry the information on the rotation of the polarization of the transmitted or reflected wave. By definition of the dielectric function $\varepsilon_{t}$ in Equation (Equation 23), one has that $\varepsilon_{t}≃\varepsilon_{ab}$ under the assumption of the small tilt angle of the planes. Then $n_{t}≃n_{b}$ and the ratios are approximately 1: one can, thus, conclude that the polarization of the transmitted or reflected wave does not differ significantly from one of the incident waves in the experiment. With the same reasoning, $\sigma_{t}≃\sigma_{ab}$, so that the transverse current is approximately parallel to the field; see Equation (Equation 3). We can conclude that $\phi ≃\phi_{J}$.

In an experiment, the measured quantity (see Appendix A) is either the transmissivity, as follows:(28)$$\begin{array}{c} T=T_{b}\cos^{2}\phi +T_{t}\sin^{2}\phi , \end{array}$$
or, analogously, the reflectivity, as follows:(29)$$\begin{array}{c} R=R_{b}\cos^{2}\phi +R_{t}\sin^{2}\phi . \end{array}$$

From these quantities, one can define the measured transverse conductivity. Indeed, under the assumption of film-thickness *d*, which is much smaller than the wavelength of the radiation inside the material and its penetration depth, one finds the following [36]:(30)$$\begin{array}{c} \sigma \left(\omega ,\eta ,\phi \right)=\frac{2}{Z_{0}d}\left(\frac{1}{T}-1\right), \end{array}$$
where $Z_{0}=4\pi /c$ is the impedance of free space. This proportionality establishes the link between the measured quantity and the theoretical conductivity we are looking for. Moreover, regarding cuprates, one can numerically estimate T≪1 in Equation (Equation 28), and then approximate σ∝1/T. Since $\sigma_{ab}\propto 1/T_{b}$ and $\sigma_{t}\propto 1/T_{t}$, from (Equation 28), one can express the measured transverse conductivity as follows:(31)$$\begin{array}{c} \sigma \left(\omega ,\eta ,\phi \right)≃\frac{\sigma_{ab}\sigma_{t}}{\sigma_{ab}\sin^{2}\phi +\sigma_{t}\cos^{2}\phi }. \end{array}$$

With $\sigma_{t}$ from Equation (Equation 4) and using the expressions of $\sigma_{ab}$ and $\sigma_{c}$ for the superconductor, one finds that the real part of the conductivity has a peak at a resonance frequency $\omega_{r}\left(\eta ,\phi \right)$ that depends on both η and ϕ:(32)$$\begin{array}{c} \omega_{r}^{2}\left(\eta ,\phi \right)=\frac{\omega_{ab}^{2}\sin^{2}\eta \sin^{2}\phi +\omega_{c}^{2}\left(1-\sin^{2}\eta \sin^{2}\phi \right)}{1-\sin^{2}\eta \cos^{2}\phi +{\left(\frac{\omega_{c}}{\omega_{ab}}\right)}^{2}\sin^{2}\eta \cos^{2}\phi }. \end{array}$$

In Figure 3b, we show the real part of Equation (Equation 30) as a function of the external polarization angle and we compare the peak in the measured conductivity with Equation (Equation 32). Indeed, the approximated expression (Equation 31) provides an excellent description of the experimental data in Refs. [33,34,35,36], and the frequency Equation (Equation 32) establishes a link between the peak of the experimental conductivity and the plasma frequencies $\omega_{ab}$ and $\omega_{c}$, which can then be extracted as fitting parameters given the angles η and ϕ. In Figure 3c, we fit experimental data from Ref. [36] to provide an estimate of the in-plane and out-of-plane plasma frequencies of the overdoped ${\mathrm{La}}_{1.87}{\mathrm{Ce}}_{0.13}{\mathrm{CuO}}_{4}$ ($T_{c}=21$ K) at 5 K. We clarify that the experimental fit is not meant to draw any conclusion on the symmetry of the superconducting order parameter, which is still debated in the context of electron-doped cuprates [36,52]. Equation (Equation 9), which is our starting point, can be derived from a microscopic model, which admits a modulation of the superconducting gap [19]. At the level of Equation (Equation 9), the gap symmetry influences the temperature dependence of the superfluid stiffnesses $D_{ab}$ and $D_{c}$, which are primarily controlled by quasiparticle excitations, while barely affecting the charge compressibility $\kappa_{0}$. In addition, the gap symmetry can impact the quasiparticle damping of the plasmon, controlling the phenomenological broadening γ, even though other mechanisms can determine its value independently of the gap symmetry. However, once the proper gap symmetry is embedded in the plasma frequencies $\omega_{ab}$ and $\omega_{c}$, the structure of the modes in Equation (Equation 18) is general.

So far, it has been empirically observed in Ref. [33] that the data could be well fitted using an effective conductivity $\sigma_{t}\left(\omega ,\eta_{\mathrm{eff}}\right)$ having the same functional form as Equation Equation (Equation 4), but with an effective tilt angle $\eta_{\mathrm{eff}}=\eta \sin\phi$. This result follows from Equation (Equation 31) in the case of a small angle η between the momentum and the *c*-axis of the crystal, which is indeed the configuration of Ref. [33]. In this case, the frequency of the peak in Equation (Equation 32) can be approximated as follows:(33)$$\begin{array}{c} \omega_{r}^{2}\left(\eta ,\phi \right)≃\omega_{ab}^{2}\sin^{2}\eta_{\mathrm{eff}}+\omega_{c}^{2}\cos^{2}\eta_{\mathrm{eff}}=\omega_{l}^{2}\left(\eta_{\mathrm{eff}}\right), \end{array}$$
where again $\eta_{\mathrm{eff}}=\eta \sin\phi$.

At first, one might consider the configuration where the THz pulse is incident at a small angle on a *c*-axis grown sample. However, computing the Fresnel conditions in this scenario results in featureless transmissivity and reflectivity, and no peak appears in the real part of the conductivity (see Appendix A for details).

## 4. Discussion

In this manuscript, we studied the optical absorption in layered superconductors within a tilted geometry, where light propagates inside the sample by forming a small angle with the stacking direction. We demonstrated that such a geometry enables observation with optics—a fundamentally zero-momentum probe—of a direct signature of the plasmon dispersion at momenta of the order of a fraction of the Brillouin zone, which is typically probed by RIXS or EELS. The basic physical mechanism behind this observation is the intrinsic mixing between transverse and longitudinal electromagnetic modes in a layered material due to the anisotropy between the in-plane and out-of-plane response. Such mixing, which is absent when light propagates along the main crystallographic axes, leads to the emergence of an absorption peak in the transverse optical conductivity in tilted geometry. Interestingly, we can analytically show that the peak frequency moves as a function of the tilting angle according to the functional law that the physical longitudinal plasmon displays at momenta larger than the scale where transverse/longitudinal mixing is relevant. In cuprates, where the SC c-axis plasmon is weakly affected by Landau damping due to the opening of a large spectral gap below $T_{c}$, the peak is well-defined at a small tilting angle, and it has indeed been observed in several electron-doped cuprates [33,34,35,36]. Here, we argue that the same effect can be seen in any layered sample, provided that the appropriate Fresnel geometry is implemented. In addition, we provide an analytical expression for the peak frequency as a function of both tilting angle and light polarization, which can be used to derive from a single set of measurements the relevant scales for plasma excitations in these systems. It is worth stressing that in recent years, after charged plasmons were detected for the first time with high-resolution RIXS [25,26,27,28,29] and EELS [30,31,32] experiments, an intense discussion emerged on the nature of charge fluctuations in these correlated materials [31,32]. The all-optical measurement proposed here is in principle a bulk probe; it is not affected by the lack of sensitivity at small momenta associated with plasmon measurements via charge-detecting probes and allows for precise control over the momentum value, which can be problematic, e.g., with EELS [32]. As a consequence, the experimental verification of this idea could provide an additional tool to explore charge fluctuations in cuprates and their possible interplay with other collective modes of the systems.

## Figures and Tables

**Figure 1 nanomaterials-14-01021-f001:**
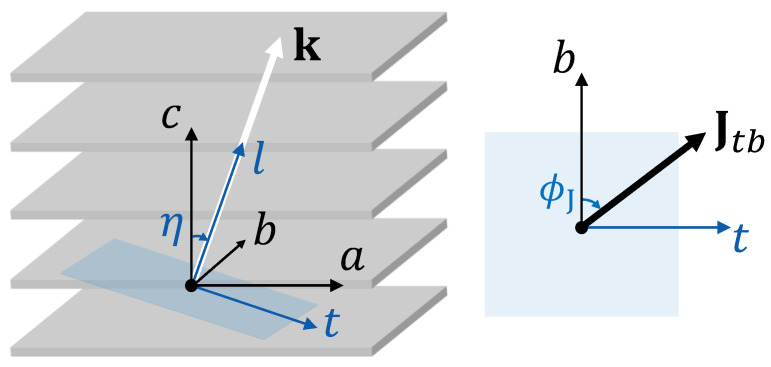
Sketch of the notation used in the manuscript to define the reference frames. The crystalline orientation defines the frame (a,b,c), and the direction of the momentum defines (t,b,l). The angles η and $\phi_{J}$ are also represented. The tb-plane is highlighted in blue.

**Figure 2 nanomaterials-14-01021-f002:**
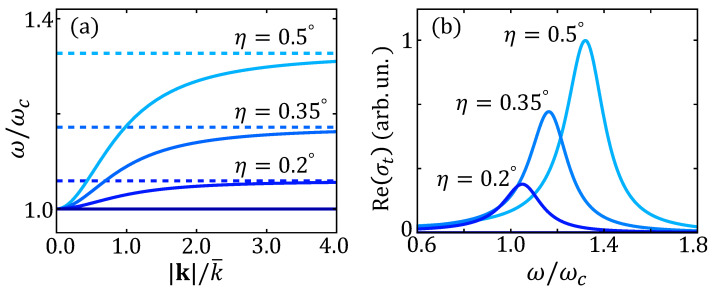
(**a**) Dispersion of the Josephson plasma mode $\omega_{-}\left(k\right)$ (solid lines) and $\omega_{l}\left(\eta \right)$ (dashed lines) for different small propagation angles, having chosen $\omega_{ab}/\omega_{c}=100$. (**b**) The real part of the conductivity $\sigma_{t}$ in the case of superconducting plasma modes for corresponding values of η of panel (**a**). The conductivity spectra are normalized to the maximum value of the peak at $\eta =0.5^{\circ }$. The phenomenological damping parameter is taken as $\gamma =0.1\omega_{c}$.

**Figure 3 nanomaterials-14-01021-f003:**
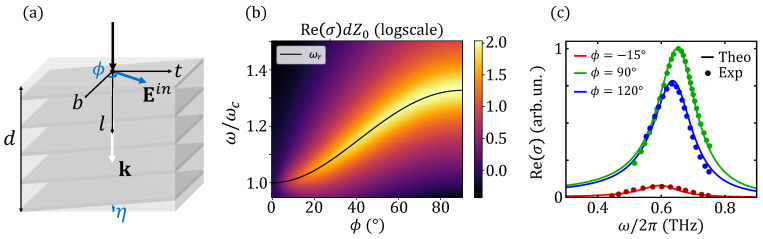
(**a**) Sketch of the experimental configuration with a THz wave at normal incidence on a tilted-grown sample of thickness *d*. The reference frame (t,b,l) for this configuration is also shown to highlight the direction of the external electric field $E^{in}$ that defines the polarization angle ϕ. (**b**) The real part of the measured conductivity as a function of frequency and the polarization angle as in Equation (Equation 30). The solid black line corresponds to $\omega_{r}\left(\eta ,\phi \right)$ as in Equation (Equation 32). In this plot, $\eta =0.25^{\circ }$, d=0.150μm, $\omega_{ab}/\omega_{c}=100$ and $\gamma =0.1\omega_{c}$. (**c**) The fit of experimental data from Ref. [36] with σ(ω,η,ϕ) for different polarization angles. Fitting parameters are extracted at once from the following measurements: $\omega_{ab}/2\pi =60\;\mathrm{THz}$, $\omega_{c}/2\pi =0.6\;\mathrm{THz}$, $\eta =0.26^{\circ }$, γ=0.075THz.

## Data Availability

Data are contained within the article.

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
