# Peer review of "Optical Absorption in Tilted Geometries as an Indirect Measurement of Longitudinal Plasma Waves in Layered Cuprates"

_nanomaterials, 2024, doi:10.3390/nano14121021_

Round 1

Reviewer 1 Report

Comments and Suggestions for Authors

The authors consider the important problem of optical conductivity in layered materials. This is especially important for the high-Tc copper and iron bsed superconductors where the layered structure obscure direct observation of the in-plane response. Here the optical conductivity and the corresponding frequencies are calculated via two methods: classical and quantum mechanical based on the path integration. Such an approach, together with the detailed description of the possible experimental geometries gives credit to the obtained expressions. Novel result here is the explicit demonstration of the entenglement between the longitudinal and transverse components that evidently affects the resulting measured quantity. I recommend the manuscript for publication in Nanomaterials. The only issue that I see is the unusual abbreviation for the plural case of "Eq." and "Fig.": instead of commenly used "Eqs." and "Figs." the authors used "Eq.s" and "Fig.s".

Author Response

We thank the Referee for the careful reading of the manuscript and for recommending publication. The plural abbreviations Eq.s and Fig.s have been corrected into Eqs. and Figs. in the revised manuscript.

Reviewer 2 Report

Comments and Suggestions for Authors

Comments on the Quality of English Language

no

Reviewer 3 Report

Comments and Suggestions for Authors

The paper mentions the use of theoretical models to compare with existing experimental data. It is suggested that the authors may conduct more experiments or use different experimental techniques to verify the predictions of the theoretical model in order to enhance the reliability and generality of the conclusions. 

This paper focuses on the optical absorption properties of superconductors under inclined geometry. It is suggested that the authors may consider exploring other types of superconductors or superconductors with different layered structures to test the applicable scope and limitations of the theoretical model.

Although the paper provides a theoretical analysis of longitudinal plasma waves in layered superconductors, the physical mechanisms behind them, especially the connection to experimental observations, can be further explored. It is suggested that the authors may discuss in more detail the effects of different physical mechanisms on the observed optical absorption peaks and how these mechanisms relate to existing physical theories.

Comments on the Quality of English Language

The English is easy to read, but the MS could bear one more proofreading to catch a scattering of minor errors.

Round 2

Reviewer 2 Report

Comments and Suggestions for Authors

The authors answered my questions and points of discussion adequately.

Reviewer 3 Report

Comments and Suggestions for Authors

This work can be considered.